# UnVELO: Unsupervised Vision-Enhanced LiDAR Odometry with Online Correction

**DOI:** 10.3390/s23083967

**Published:** 2023-04-13

**Authors:** Bin Li, Haifeng Ye, Sihan Fu, Xiaojin Gong, Zhiyu Xiang

**Affiliations:** Faculty of the College of Information Science and Electronic Engineering, Zhejiang University, Hangzhou 310027, China; 3130102392@zju.edu.cn (B.L.); 22031107@zju.edu.cn (H.Y.); sihan_fu@zju.edu.cn (S.F.); xiangzy@zju.edu.cn (Z.X.)

**Keywords:** visual–LiDAR odometry, deep learning, multi-sensor fusion, test time optimization

## Abstract

Due to the complementary characteristics of visual and LiDAR information, these two modalities have been fused to facilitate many vision tasks. However, current studies of learning-based odometries mainly focus on either the visual or LiDAR modality, leaving visual–LiDAR odometries (VLOs) under-explored. This work proposes a new method to implement an unsupervised VLO, which adopts a LiDAR-dominant scheme to fuse the two modalities. We, therefore, refer to it as unsupervised vision-enhanced LiDAR odometry (UnVELO). It converts 3D LiDAR points into a dense vertex map via spherical projection and generates a vertex color map by colorizing each vertex with visual information. Further, a point-to-plane distance-based geometric loss and a photometric-error-based visual loss are, respectively, placed on locally planar regions and cluttered regions. Last, but not least, we designed an online pose-correction module to refine the pose predicted by the trained UnVELO during test time. In contrast to the vision-dominant fusion scheme adopted in most previous VLOs, our LiDAR-dominant method adopts the dense representations for both modalities, which facilitates the visual–LiDAR fusion. Besides, our method uses the accurate LiDAR measurements instead of the predicted noisy dense depth maps, which significantly improves the robustness to illumination variations, as well as the efficiency of the online pose correction. The experiments on the KITTI and DSEC datasets showed that our method outperformed previous two-frame-based learning methods. It was also competitive with hybrid methods that integrate a global optimization on multiple or all frames.

## 1. Introduction

Due to the natural complementary characteristics of visual and LiDAR information, these two modalities have been fused to facilitate many computer vision tasks such as 3D object detection [1,2,3], depth completion [4,5,6], and scene flow estimation [7,8]. Especially in the deep learning era, various fusion techniques including early-, late-, or multi-stage-fusion [6] and vision- or LiDAR-dominant fusion [8] have been developed towards each specific vision task. Contrastively, in learning-based odometries, the majority of researches still focus on either visual odometry (VO) [9,10,11,12,13,14,15,16] or LiDAR odometry (LO) [17,18,19,20,21,22,23,24,25,26], leaving visual–LiDAR-fusion-based odometry under-explored.

Most previous learning-based visual–LiDAR odometries (VLOs) [27,28,29,30] commonly adopt a vision-dominant fusion scheme, which projects a LiDAR frame into a camera frame and leads to a sparse depth map. Therefore, how to deal with sparse depth maps or generate dense depth maps becomes a challenge to achieve an accurate VLO. Besides, self-supervised VLOs often employ a view synthesis loss [27,28,29] or an additional point-to-point distance loss [28,29], for learning. The former loss depends on the predicted dense depth maps, which are inevitably noisy, while the latter is sensitive to the sparsity of points.

In this work, we adopted a LiDAR-dominant fusion scheme to implement our unsupervised VLO. Specifically, the 3D point clouds of a LiDAR frame are converted into a dense vertex map via spherical projection as in LOs [17,18,19,20,21]. A vertex color map is then generated, which assigns each vertex a color retrieved from the aligned visual image. We further performed LiDAR-based point-to-plane matching within locally planar regions, while establishing pixel correspondences based on visual images for cluttered regions. A geometric consistency loss and a visual consistency loss are, respectively, defined for these two different types of regions. By this means, the complementary characteristics of the visual and LiDAR modalities are well exploited. Moreover, our LiDAR-dominant scheme does not need to predict dense depth maps, avoiding the construction of a complex depth-prediction network and preventing the noise introduced by the predicted depth. Considering that LiDAR plays the dominant role in the visual–LiDAR fusion, we named our method unsupervised vision-enhanced LiDAR odometry (UnVELO).

The losses used for UnVELO are unsupervised, requiring no ground truth labels for training. This implies that the losses can also be applied for optimization during test time. Recently, test time optimization has been explored in several unsupervised VOs [14,15,16] to either refine the weights of their networks or refine the predicted outputs further, referred to as online learning and online correction, respectively. Compared with online learning [14,16], the online correction scheme [15] significantly reduces the number of parameters to be optimized, leading to a higher computational efficiency. Thus, in this work, we adopted online correction to refine the pose predicted by the trained UnVELO network. In contrast to the optimization loss in VOs that tightly couples depth and pose prediction, our UnVELO predicts the pose only, making test time optimization more effective.

In summary, the proposed method distinguishes itself from previous self-supervised VLOs in the following aspects:We adopted a LiDAR-dominant fusion scheme to implement an unsupervised visual–LiDAR odometry. In contrast to previous vision-dominant VLOs [27,28,29], which predict both the pose and dense depth maps, our method only needs to predict the pose, avoiding the inclusion of the noise generated from the depth prediction.We placed a geometric consistency loss and a visual consistency loss, respectively, on locally planar regions and cluttered regions, by which the complementary characteristics of the visual and LiDAR modalities can be exploited well.We designed an online pose-correction module to refine the predicted pose during test time. Benefiting from the LiDAR-dominant scheme, our online pose correction is more effective than its vision-dominant counterparts.The proposed method outperformed previous two-frame-based learning methods. Besides, while introducing two-frame constraints only, our method achieved a performance comparable to the hybrid methods, which include a global optimization on multiple or all frames.

## 2. Related Work

Pose estimation is a key problem in simultaneous localization and mapping (SLAM), which plays an important role in various applications such as autonomous driving [31], 3D reconstruction [32], and augmented reality [33]. To date, most odometry methods use a visual camera or LiDAR for pose estimation. The visual camera can provide dense color information of the scene, but is sensitive to the lighting conditions, while the latter can obtain accurate, but sparse distance measurements; thus, they are complementary.

There are also some works that have attempted to exploit other on-board sensors for pose estimation. For example, radar odometry [34] adopts an extended Kalman filter to propagate the motion state from the IMU and corrects the drift by the measurements from radar and GPS. DeepLIO [19] uses two different networks to extract motion features from LiDAR and IMU data, respectively, and fuses the features by an attention-based soft fusion module for pose estimation. A discussion of the full literature of these methods is beyond the scope of this paper. In this section, we mainly focus on the methods using a visual camera and LiDAR.

### 2.1. Visual and LiDAR Odometry

Although state-of-the-art performance is maintained by conventional methods such as LOAM [32], V-LOAM [35], and SOFT2 [36], learning-based visual or LiDAR odometries have been attracting great research interest. In this work, we briefly review the learning-based methods.

#### 2.1.1. Visual Odometry

A majority of end-to-end visual odometry works focus on self- or unsupervised monocular VOs [9,10,11,37,38,39,40]. They take a monocular image sequence as the input to jointly train the pose- and depth-prediction networks by minimizing a view synthesis loss. To overcome the scale ambiguity problem in monocular VOs, SC-SfMLearner [10] and Xiong et al. [11] proposed geometric consistency constraints to achieve globally scale-consistent predictions, while UnDeepVO [41] opts to learn from stereo sequences. Besides, additional predictions such as the motion mask [41] and optical flow [42] are also integrated to address motion or occlusion problems. Recently, hybrid methods such as DVSO [43] and D3VO [44] have integrated end-to-end networks with traditional global optimization modules to boost the performance.

#### 2.1.2. LiDAR Odometry

In order to deal with sparse and irregular 3D point clouds, different representations have been used to implement learning-based LiDAR odometries. For instance, DeepLO [18], LO-Net [17], and DMLO [45] project 3D point clouds of each LiDAR frame into a 2D map via spherical or cylindrical projection, by which the well-developed 2D CNNs can be used for learning. SelfVoxeLO [24] and RLO [26] voxelize 3D point clouds and use 3D CNNs to retain the 3D topology relations. PWCLO-Net [25] and SVDLO [46] attempt to process the raw 3D point clouds by PointNet-based models [47]. For the training loss under self-supervision, ICP-like [48,49,50], point-to-point matching loss [51], point-to-plane matching loss [18,20], and plane-to-plane matching loss [20] are extensively used.

#### 2.1.3. Visual–LiDAR Odometry

In contrast to the extensive studies on VOs and LOs, the works on visual–LiDAR odometries are relatively scarce. Existing learning-based VLOs include Self-VLO [27], RGBD-VO [28], MVL-SLAM [29], and Tibebu et al. [52]. Most of them [27,28,29] project 3D points into a camera frame for depth representation. Then, visual and depth images are either concatenated or separate to feed into 2D CNNs for feature extraction and fusion. However, the feature fusion of sparse depth maps and dense visual images can be challenging. It is difficult to extract reliable multi-modal features for the areas that are not covered by the depth. Besides, their main supervision signals come from the view synthesis loss [37], which is still sensitive to lighting condition changes. Differently, Tibebu et al. [52] projected LiDAR data into a 1D vector or 2D range map within the LiDAR frame. They feed visual images and LiDAR data into two streams for feature extraction and constructed a fusion layer and an LSTM module for feature fusion. Due to the different resolution of visual and LiDAR data, they adopt two independent modules to extract visual and LiDAR features respectively, and perform the feature fusion at the last layer only. In contrast to the mentioned VLOs, our LiDAR-dominant method projects the visual and LiDAR data into two dense images with the same size and obtains multi-modal features by a single feature extractor. Moreover, our method predicts relative poses only for training; thus, the optimization is also more efficient compared to the vision-dominant methods, which need to predict both the depth and pose.

### 2.2. Visual–LiDAR Fusion

Visual–LiDAR fusion has been widely investigated in various tasks including depth completion [5,6], scene flow estimation [7,8], and visual–LiDAR odometry [27,28,29]. According to which view plays the dominant role, we classify existing fusion strategies into vision-dominant, LiDAR-dominant, or vision–LiDAR-balanced ones. Most depth completion [5,6], scene flow estimation [7], and VLOs [27,28,29] adopt the vision-dominant strategy, which projects LiDAR frames into camera frames, leading to sparse depth maps, which are hard to deal with. Some scene flow estimation [8] and VLO [52] methods adopt the vision–LiDAR-balanced fusion strategy, which constructs two streams to extract features, respectively, from the LiDAR and camera views, often along with a complex module to fuse the features of the two modalities.

In order to avoid dealing with sparse depth maps or designing complex fusion modules, we adopted the LiDAR-dominant fusion scheme. It first projects 3D LiDAR points into a dense vertex map and then colorizes each vertex with the visual information. The LiDAR-dominant fusion scheme is also adopted by several 3D object detection methods such as PointAugmenting [53]. In contrast to these works that paint 3D points [53], we encode LiDAR data as 2D vertex maps for more efficient computation.

### 2.3. Test Time Optimization

Test time optimization is a strategy to refine the weights or outputs of a trained network during test time. Recently, this strategy has been applied to various unsupervised learning tasks [16,54,55,56,57,58] since their losses require no ground truth, making test time optimization possible. In self-supervised VOs, Li et al. [14] proposed online meta-learning to continuously adapt their VO networks to new environments. Li et al. [58] optimized the predicted depth and flow via a Gauss–Newton layer and took the optimized results as pseudo labels to supervise the online learning of the depth and flow models. DOC [15] designs a deep online correction framework to efficiently optimize the pose predicted by a trained VO. GLNet [16] adopts both weight and output fine-tuning modules to boost its performance.

All the above-mentioned self-supervised VOs [14,15,16] use the view synthesis loss for learning and test time optimization. This loss involves the depth and pose, both of which are predicted by the trained networks. Therefore, the quality of their pose refinement is affected by the predicted depth maps, which are noisy. In this work, we applied online correction to refine the pose predicted by our UnVELO network during test time. In contrast to VOs [14,15,16], our losses only involve the predicted pose while using depth information directly converted from accurate LiDAR measurements. It therefore facilitates the pose refinement.

## 3. Materials and Methods

As shown in Figure 1, the proposed method consists of the data pre-processing, pose estimation, and online correction modules. Given two consecutive LiDAR scans $\langle S_{t},S_{t+1}\rangle$ and synchronized visual images $\langle I_{t},I_{t+1}\rangle$, our method first generates the corresponding vertex maps, normal maps, and vertex color maps in the data pre-processing step. Then, the vertex maps and vertex color maps are concatenated and input into a network for pose estimation. During test time, the pose predicted from the network is further optimized via the online pose-correction module (OPC).

### 3.1. Data Pre-Processing

#### 3.1.1. Vertex Map

As is common practice [18], we adopted a spherical projection $\pi \left(\cdot \right):R^{3}\mapsto R^{2}$ to convert each 3D point in a LiDAR frame into a pixel on a vertex map. Specifically, a 3D point $p={\left[p_{x},p_{y},p_{z}\right]}^{T}$ within a field of view (FOV) is projected into a 2D pixel u via
(1)$$u=\left[\begin{array}{c} round\left(\left(f_{h}/2-\arctan\left(p_{y},p_{x}\right)\right)/\delta_{h}\right)\\ round\left(\left(f_{vu}-\arctan\left(p_{z},\sqrt{p_{x}^{2}+p_{y}^{2}}\right)\right)/\delta_{v}\right) \end{array}\right],$$
in which $f_{h}$ is the horizontal FOV and $f_{vu}$ is the upper part of the vertical FOV $f_{v}$. Moreover, $\delta_{h}$ and $\delta_{v}$ denote the horizontal and vertical angular resolutions, respectively. We then define a vertex map $V\in R^{H\times W\times 3}$ with $H=\left\lfloor f_{v}/\delta_{v}\right\rfloor$ and $W=\left\lfloor f_{h}/\delta_{h}\right\rfloor$. If there is a 3D point p projected to the pixel u, we define V(u)=p; otherwise, V(u)=0. Thus, along with the vertex map, we also obtain a binary mask Mv to indicate the black pixels as follows:(2)Mv(u)=1(||V(u)||>0),
where 1(·) is an indicator and ||·|| denotes the $L_{2}$ norm.

#### 3.1.2. Normal Map

Normal vectors are important for point cloud registration as they can characterize the local geometry around points [49,50]. In this work, we adopted singular-value decomposition (SVD) [50,59] to estimate the normals. For each pixel u and its associated point p=V(u), we compute the mean μ and covariance Σ within a neighboring set $N_{p}$ as follows:(3)$$\begin{array}{ccc} \mu & = & \frac{1}{|N_{p}|}\sum_{p_{i}\in N_{p}}p_{i}, \end{array}$$(4)$$\begin{array}{ccc} \Sigma & = & \frac{1}{|N_{p}|}\sum_{p_{i}\in N_{p}}{\left(p_{i}-\mu \right)}^{T}\left(p_{i}-\mu \right), \end{array}$$
where |·| denotes the cardinality of a set. Empirically, we set $N_{p}=\{p_{i}\left|\;|\right|p_{i}-p||<0.15||p||\wedge p_{i}\in W\}$ as the set of neighboring points near p, and W is a local window of size 5×7 centered at u on the vertex map.

Then, we obtain the singular vector n corresponding to the minimum singular value of Σ by SVD and take it as the normal vector. The normal map $N\in R^{H\times W\times 3}$ is then defined by N(u)=n for valid pixels and N(u)=0 otherwise. We generate a confidence map C by computing the similarity of the normals with four neighbors [18]. That is,
(5)$$C\left(u\right)=\sum_{u_{i}\in N_{u}}\frac{1+N\left(u_{i}\right)\cdot N\left(u\right)/||N\left(u_{i}\right)||||N\left(u\right)\left|\right|}{8},$$
where · is the inner product and $N_{u}$ denotes the four connected neighboring pixels of u. The confidence is in [0,1]. A high confidence indicates a planar surface, and a low confidence often corresponds to a cluttered region. Figure 2 shows that the regions on the ground and walls have high confidence and those on the object boundaries or plants have low confidence. We therefore generate a binary mask Mn to indicate locally planar regions by
(6)$$\mathrm{Mn}\left(u\right)=1(|N\left(u\right)|>0\wedge C\left(u\right)>\delta_{c}),$$
where $\delta_{c}$ is the threshold of the confidence.

#### 3.1.3. Vertex Color Map

In order to fuse visual information, we generate a vertex color map $\mathrm{Vc}\in R^{H\times W\times 3}$ based on the vertex map V and a synchronized visual image I. Specifically, for each vertex p=V(u) on the vertex map, we retrieve the color of its corresponding pixel $u^{\prime }$ at the visual image through the following camera projection:(7)$$\left[\begin{array}{c} u^{\prime }\\ 1 \end{array}\right]={\left[K|0\right]}_{3\times 4}T_{C\leftarrow L}\left[\begin{array}{c} p\\ 1 \end{array}\right],$$
where $K\in R^{3\times 3}$ denotes a camera intrinsic matrix and $T_{C\leftarrow L}\in R^{4\times 4}$ is the transformation matrix from the LiDAR to the camera coordinates. Since the values of the projected $u^{\prime }$ are continuous, we obtain its color via the bilinear sampling scheme as in VOs [37]. That is to say, $\tilde{I}\left(u^{\prime }\right)=\sum_{u_{i}^{\prime }\in N_{u^{\prime }}}\omega_{i}I\left(u_{i}^{\prime }\right)$, in which $N_{u^{\prime }}$ contains the four closest pixels of $u^{\prime }$, $\omega_{i}$ is linearly proportional to the spatial proximity between $u^{\prime }$ and $u_{i}^{\prime }$, and $\sum \omega_{i}=1$. Then, we obtain a vertex color map $\mathrm{Vc}\left(u\right)=\tilde{I}\left(u^{\prime }\right)$, along with a binary mask Mc:(8)Mc(u)=1(|Vc(u)|>0).

### 3.2. Pose Estimation

#### 3.2.1. Network Architecture

As shown in Figure 3, we constructed a fully convolutional neural network composed of a feature encoder and a pose estimator to infer the relative pose between two consecutive frames. Two consecutive vertex maps $\langle V_{t},V_{t+1}\rangle$ and their corresponding vertex color maps $\langle {\mathrm{Vc}}_{t},{\mathrm{Vc}}_{t+1}\rangle$ are concatenated as the input, which has a size of H×W×12. The feature encoder contains 13 convolutional layers, where the kernel size of the first layer is 5×5 and the rest are 3×3. The vertical and horizontal strides of Layers 2,6,10 were set to (1,2), the strides of Layer 4,8 to (2,2), and the remaining with a stride of (1,1). This implies that only 2 down-sampling operations are performed in the vertical direction, but 5 down-sampling operations are performed in the horizontal direction, since the input’s width is greater than its height. The pose estimator predicts a 3D translation vector ${\left[t_{x},t_{y},t_{z}\right]}^{T}$ and a 3D Euler vector ${\left[r_{x},r_{y},r_{z}\right]}^{T}$ through two separate branches. Finally, we obtain a 6D vector $P_{t\leftarrow t+1}={\left[t_{x},t_{y},t_{z},r_{x},r_{y},r_{z}\right]}^{T}$, from which a 4×4 transformation matrix $T_{t\leftarrow t+1}$ is constructed.

#### 3.2.2. Training Loss

We designed a loss L composed of a geometric loss $L_{geo}$ and a visual loss $L_{vis}$ to train the pose-estimation network. That is,
(9)$$L=L_{geo}+\lambda L_{vis},$$
where λ is a scaling factor to balance two terms. The details are introduced in the following.

The geometric loss $L_{geo}$ places geometric constraints on locally planar regions where the normals have high confidence. We adopted the point-to-plane distance [18] to measure the registration error of points in two LiDAR frames. Formally, given an estimated pose $P_{t\leftarrow t+1}$ and the corresponding transformation matrix $T_{t\leftarrow t+1}$ from frame t+1 to frame *t*, for each pixel $u_{t+1}$ in frame t+1, we transform the corresponding 3D point $p_{t+1}=V_{t+1}\left(u_{t+1}\right)$ into frame *t* by
(10)$$\left[\begin{array}{c} p_{t}^{\prime }\\ 1 \end{array}\right]=T_{t\leftarrow t+1}\left[\begin{array}{c} p_{t+1}\\ 1 \end{array}\right],$$
and obtain its registered correspondence ${\hat{p}}_{t}$ in frame *t* by using a line-of-sight criterion [50]. That is,
(11)$${\hat{p}}_{t}=V_{t}\left(\pi \left(p_{t}^{\prime }\right)\right).$$Then, the confidence-weighted point-to-plane distance at pixel $u_{t+1}$ is defined as follows:(12)$$d_{geo}\left(u_{t+1}\right)=C_{t}\left(\pi \left(p_{t}^{\prime }\right)\right)\left|N_{t}\left(\pi \left(p_{t}^{\prime }\right)\right)\cdot \left(p_{t}^{\prime }-{\hat{p}}_{t}\right)\right|.$$

The geometric loss is further defined by
(13)$$L_{geo}=\frac{1}{|M_{geo}|}\sum_{u_{t+1}}d_{geo}\left(u_{t+1}\right)M_{geo}\left(u_{t+1}\right),$$
(14)$$M_{geo}={\mathrm{Mv}}_{t+1}⊙{\mathrm{Mn}}_{t+1}.$$Here, ⊙ denotes the elementwise multiplication. $M_{geo}$ is a binary mask to select valid and highly confident pixels on locally planar regions.

The visual loss $L_{vis}$ enforces visual consistency for pixels on object boundaries and cluttered regions. This loss is complementary to the geometric loss. In contrast to the geometric loss focusing on planar regions that have reliable normals, but often lack texture, the visual loss pays attention to the regions with less confident for the normals, but textured. Specifically, we transform $V_{t+1}$ into frame *t* via Equation (10) and generate a new vertex color map $\tilde{V}c_{t}$ together with its mask $\tilde{M}c_{t}$ from the transformed v. We then adopted the photometric error to measure the difference of the corresponding pixels in two vertex color maps, that is
(15)$$d_{vis}\left(u_{t+1}\right)=\left|{\mathrm{Vc}}_{t+1}\left(u_{t+1}\right)-\tilde{V}c_{t}\left(u_{t+1}\right)\right|,$$
and the visual loss is defined as follows:(16)$$L_{vis}=\frac{1}{|M_{vis}|}\sum_{u_{t+1}}d_{vis}\left(u_{t+1}\right)M_{vis}\left(u_{t+1}\right),$$
(17)$$M_{vis}=\tilde{M}c_{t}⊙{\mathrm{Mc}}_{t+1}⊙\left(1-{\mathrm{Mn}}_{t+1}\right).$$Here, $M_{vis}$ marks the valid pixels with low confidences of the normals, which are often lying on object boundaries and cluttered regions.

### 3.3. Online Pose Correction

#### 3.3.1. Formulation

In the training of the pose-estimation network that is parameterized by Θ, the model in essence performs the following optimization:(18)$$\Theta^{*}=\underset{\Theta }{\mathrm{argmin}}\sum_{i\in TS}L\left(V_{t}^{i},V_{t+1}^{i},{\mathrm{Vc}}_{t}^{i},{\mathrm{Vc}}_{t+1}^{i},P_{t\leftarrow t+1}^{i}\left(\Theta \right)\right).$$Here, L is the loss defined in Equation (9) and TS denotes the training set. Note that all parameters of the loss are ignored in Equation (9) for the sake of brevity.

Once the network is trained, the network’s parameters $\Theta^{*}$ are fixed for inference. During test time, when two consecutive frames are given, we further optimized the predicted pose $P_{t\leftarrow t+1}$ while keeping $\Theta^{*}$ fixed. This online correction benefits from the unsupervised loss, which requires no ground truth labels. The optimization is conducted by
(19)$$P_{t\leftarrow t+1}^{*}=\underset{P_{t\leftarrow t+1}}{\mathrm{argmin}}L\left(V_{t},V_{t+1},{\mathrm{Vc}}_{t},{\mathrm{Vc}}_{t+1},P_{t\leftarrow t+1}\left(\Theta^{*}\right)\right),$$
which can be solved by the gradient descent method while taking the pose predicted by the network as the initial value. We adopted Adam [60] to minimize it for *N* iterations.

#### 3.3.2. Hard Sample Mining

Hard sample mining (HSM) is widely used in deep metric learning [61] and person re-identification [62] to speed up convergence and improve the learned embeddings. In this paper, we took HSM as a plug-and-play component in the OPC to filter the easy samples and outliers for optimization, and thus, we can focus on the challenging correspondences to facilitate the convergence. The sampling metric is defined within a neighborhood of each sample. More specifically, given a point $p_{t+1}=V_{t+1}\left(u_{t+1}\right)$, we take all neighboring points of its correspondence ${\hat{p}}_{t}$ obtained by Equation (11) as matching candidates and calculate their matching errors. Then, the relative standard deviation (RSD) of these matching errors is calculated. A point having a large RSD implies that either the mean of the matching errors within the neighborhood is small (i.e., easy point samples) or the standard deviation of the errors is large (i.e., outlier points). Therefore, we leave out both easy samples and outliers, while selecting the remaining as hard samples. That is,
(20)$$M_{geo}^{hard}=1\left(\mathrm{RSD}\left(u_{t+1}\right)<mean\left(\mathrm{RSD}\left(u_{t+1}\right)\right)\right).$$Then, we update the binary mask $M_{geo}$ by
(21)$$M_{geo}=M_{geo}⊙M_{geo}^{hard},$$
and all the others in the optimization loss are kept unchanged.

Figure 4 presents two examples of our hard sample mining results. It shows that a portion of points on locally planar regions and most points on trees are selected. When only taking the selected points into consideration for the geometric loss, the online pose correction procedure can be facilitated.

## 4. Results

### 4.1. Experimental Settings

#### 4.1.1. Dataset and Evaluation Metrics

We evaluated the proposed method on the KITTI odometry benchmark [63] and the DSEC dataset [64]. KITTI [63] contains 22 sequences of LiDAR scans captured by a Velodyne HDL-64E, together with synchronized visual images. The ground truth pose of sequence 00-10 is provided for training or evaluation. DSEC [64] contains images captured by a stereo RGB camera and LiDAR scans collected by a Velodyne VLP-16 LiDAR. However, since the LiDAR and camera were not synchronized, we took the provided ground truth disparity maps to obtain the 3D point clouds. Moreover, as no ground truth pose was available, we took the pose estimated by LOAM [32] as the pseudo ground truth for the evaluation. For the performance evaluation, following the official criteria provided in the KITTI benchmark, we adopted the average translational error (%) and rotational error (deg/100 m) on all possible sub-sequences of length (100, 200, ⋯, 800) meters as the evaluation metrics.

#### 4.1.2. Implementation Details

The propose method was implemented in PyTorch [65]. For the data pre-processing on KITTI, we set $f_{h}=80^{\circ }$ considering the camera’s horizontal FOV, $f_{v}=24^{\circ }$, and $f_{vu}=3^{\circ }$. In addition, we set $\delta_{h}=0.375$ and $\delta_{v}=0.1786$ in order to generate vertex maps with a size of 64×448. Note that only 3D points within the camera’s FOV were projected to ensure most vertexes can be associated with visual information. Besides, each visual image was resized into 192×624 for computational efficiency. For data pre-processing on DSEC, we set $f_{h}=52^{\circ }$, $f_{v}=28^{\circ }$, and $f_{vu}=10^{\circ }$. $\delta_{h}$ and $\delta_{v}$ were set according to the generated vertex maps with a size of 64×320. We used images captured by the left RGB camera and cropped the images to 840×1320 from the top-right corner to ensure that they were covered by the ground truth disparities. The cropped images were further resized to 384×608 for computational efficiency.

In the training stage, we used the Adam optimizer [60] with $\beta_{1}=0.9$, $\beta_{2}=0.999$ and a mini-batch size of four to train the model for 300 K iterations. In KITTI, the initial learning rate starts from 0.0001 and decreases by 0.8 every 30 K iterations. The scalar λ in the training loss was set to 1.0 empirically, and the confidence threshold $\delta_{c}=0.9$. For the online pose correction, we also adopted Adam with $\beta_{1}=0.9$ and $\beta_{2}=0.999$. Considering the difference of the translation and rotation values, we set the learning rate of the translation parameters as 0.025, while the learning rate of the rotation as 0.0025. In DSEC, the initial learning rate was set to 0.0002. The learning rate of the translation and rotation parameters in the online pose correction were set to 0.02 and 0.001. All other parameters were kept the same as those in KITTI.

### 4.2. Ablation Studies

We first conducted a series of experiments on KITTI to investigate the effectiveness of each component proposed in our pose-estimation network. To this end, we checked the following model variants: (1) UnVELO1: only vertex maps were input to the pose network, and only the geometric loss was used for training; (2) UnVELO2: both vertex maps and vertex color maps were input, but only the geometric loss was used; (3) UnVELO3: both vertex maps and vertex color maps were input, while both the geometric and visual loss were used for training. All these model variants were tested without the online pose-correction module. In order to illustrate the advantage of the LiDAR-dominant fusion (LDF) scheme, a vision-dominant-fusion (VDF)-based model VLO [27] was also included for comparison. (Note that this VLO model corresponds to the basic model “VLO1” in [27]. The full model “VLO4” in [27] was not taken since it adopts additional data augmentation and flip consistency constraints, while we wanted to keep the setting the same as our UnVELO models.) Besides, as is common practice [15,18,24,37,66], we took sequence 00-08 for training and sequence 09-10 for testing. The results are presented in Table 1. We observed that the input of both vertex maps and vertex color maps can slightly improve the performance, and the use of both the geometric and visual loss boosted the performance further. Moreover, UnVELO3 outperformed the VLO model by a considerable margin, indicating the advantage of the LiDAR-dominant scheme.

We further investigated the effectiveness of the online correction (OC) module. In this experiment, different numbers of iterations and the model without hard sample mining (w/o HSM) were tested, and their results are reported in Table 2. In addition, we applied the online pose-correction scheme to the vision-dominant model VLO and report their results for comparison. Since VLO predicts both the depth and pose, we also tested the model that refines both the pose and depth in the online correction, denoted as “VLO+OC-40 Opt-Dep”. The experiments showed that the online correction improved the performance for both VLO and UnVELO. In VLO, the results of “VLO+OC-40 Opt-Dep” indicated that the additional depth refinement made the online correction harder, and the performance may degenerate. When comparing UnVELO with VLO, we saw that the online correction was much more effective for our UnVELO model and the performance was consistently improved with the increase of the iteration number.

Figure 5 plots the performance when varying with the iteration number of the online correction. It shows that both translation and rotation errors converged around 40 iterations and the hard sample mining scheme led to a more stable convergence and better performance with a minor increase in the runtime (about 1 ms per iteration).

### 4.3. Runtime Analysis

We conducted all experiments on a single NVIDIA RTX 2080Ti GPU and provide the runtime of each model variant in Table 2. When comparing the UnVELO and VLO model variants with the same number of online correction iterations, we observed that the LiDAR-dominant UnVELO models performed more efficiently than the visual-dominant counterparts. Moreover, the UnVELO model was very efficient when online correction was not applied. However, as expected, the runtime of our model went up along with the increase of the iteration number. For compensation, we additionally conducted an experiment that performed the online correction on frame *t* and frame t+2, which is denoted as the “UnVELO3+OC-40-Inter2” model. It achieved comparable results to the one conducting the online correction on two consecutive frames while taking half the time.

### 4.4. Comparison to State-of-the-Art

#### 4.4.1. Comparison on KITTI

Finally, we compared our full model, which is “UnVELO3+OC-40” in Table 2, but is referred to as UnVELO here, with state-of-the-art learning-based methods on the KITTI odometry benchmark. Since some methods [15,18,20,27,66,67] are trained on sequence 00-08 and tested on sequence 09-10, while the others [17,21,24,25,26,45,46,68] are trained on sequence 00-06 and evaluated on sequence 07-10, we conducted two experiments following these two different splittings. The results are respectively presented in Table 3 and Table 4. The results showed that our method outperformed most end-to-end unsupervised odometry methods in both experimental settings. It was also comparable to hybrid methods, which integrate global optimization modules.

In addition, Figure 6 plots the trajectories obtained by our UnVELO and the code-available methods including SeVLO [27], DeepLO [18], and DeLORA [20] for a more intuitive comparison. The plots show that our method obtained trajectories closer to the ground truth than the others.

#### 4.4.2. Comparison on DSEC

We also compared the proposed method with SeVLO [27], DeepLO [18], and DeLORA [20] on the DSEC dataset, which contains both day and night scenarios with large illumination variations. We should note that the generated vertex map will be sparser under poor illumination conditions, since the 3D point clouds are obtained from the disparity maps, as shown in Figure 7. Besides, the LiDAR used in this dataset had only 16 lines, which is too sparse for the LOs. Thus, the compared LOs also took the 3D point clouds obtained from the disparity maps as the input.

For a more comprehensive comparison, we chose a day sequence “zurich_city_06_a” and a night sequence “zurich_city_03_a” with totally different illumination conditions for testing. The comparison results are presented in Table 5, and the trajectories are plotted in Figure 8. The results demonstrated the effectiveness of our method.

## 5. Conclusions

Vision-dominant VLOs need to predict both dense depth maps and relative poses for unsupervised training, but the noisy predicted depth map limits the accuracy of the predicted pose. In this paper, we proposed an unsupervised vision-enhanced LiDAR odometry, which projects visual and LiDAR data into two dense images with the same resolutions to facilitate the visual–LiDAR fusion. A geometric loss and a visual loss were proposed to exploit the complementary characteristics of these two modalities, leading to a better robustness to the lighting condition variations compared to the vision-dominant VLOs trained with view the synthesis loss. Moreover, an online correction module was also designed to refine the predicted pose during test time. The experiments on KITTI and DSEC showed that our method outperformed the other two-frame-based learning methods and was even competitive with hybrid methods. Besides, while the pose accuracy of vision-dominant VLOs is limited by the noisy predicted dense depth, our LiDAR-dominant method only needs to predict the pose, which not only achieved better performance, but also improved the optimization efficiency significantly.

Our method provides a novel promising design for VLO. In future work, it will be necessary to explore the long-term temporal constraints for pose correction to improve the robustness to the abrupt motion changes, dynamic objects, and other disturbances.

## Figures and Tables

**Figure 1 sensors-23-03967-f001:**
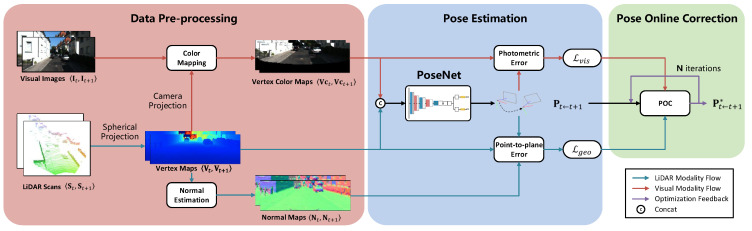
An overview of the proposed method. It consists of the data pre-processing, pose estimation, and online pose-correction modules. Given two consecutive LiDAR scans $\langle S_{t},S_{t+1}\rangle$ and visual images $\langle I_{t},I_{t+1}\rangle$, the data pre-processing step produces vertex maps $\langle V_{t},V_{t+1}\rangle$, normal maps $\langle N_{t},N_{t+1}\rangle$, as well as vertex color maps $\langle {\mathrm{Vc}}_{t},{\mathrm{Vc}}_{t+1}\rangle$. The vertex maps and vertex color maps are concatenated and fed into a pose-estimation network to predict the pose $P_{t\leftarrow t+1}$ from frame t+1 to frame *t*. During test time, the predicted pose is further optimized via the online pose-correction module.

**Figure 2 sensors-23-03967-f002:**
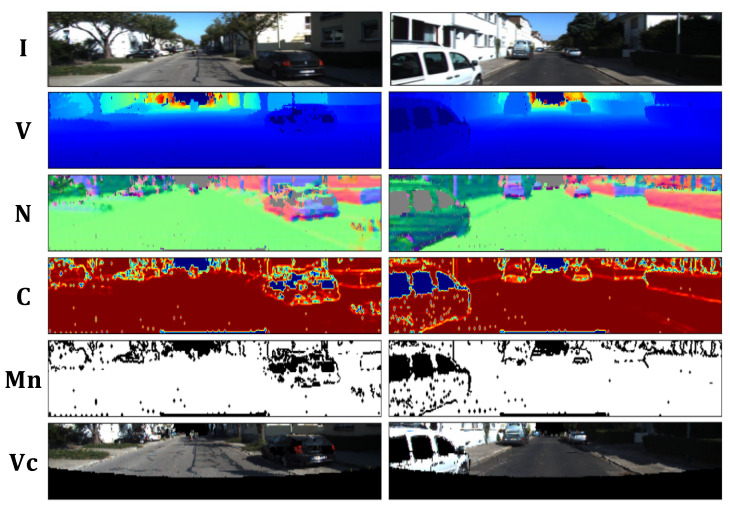
Typical examples in sequence 00 of KITTI. The visual image I, vertex map V, normal map N, normal confidence map C, normal binary mask Mn, and vertex color map Vc are presented from top to bottom.

**Figure 3 sensors-23-03967-f003:**
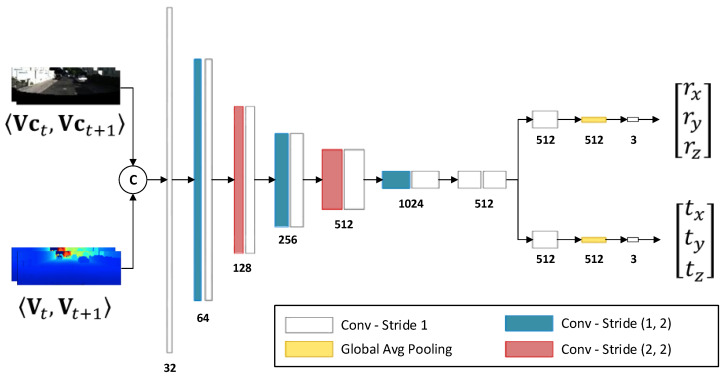
The architecture of our pose-estimation network.

**Figure 4 sensors-23-03967-f004:**
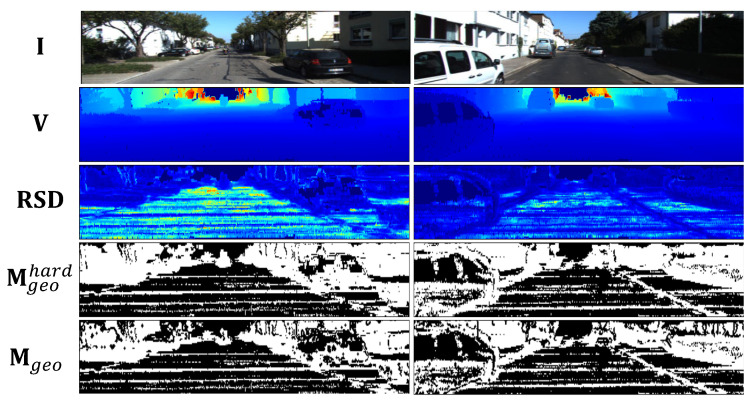
Typical examples in sequence 00 of KITTI. The visual image I, vertex map V, relative standard deviation (RSD), hard sample mining result $M_{geo}^{hard}$, and the updated $M_{geo}$ are presented.

**Figure 5 sensors-23-03967-f005:**
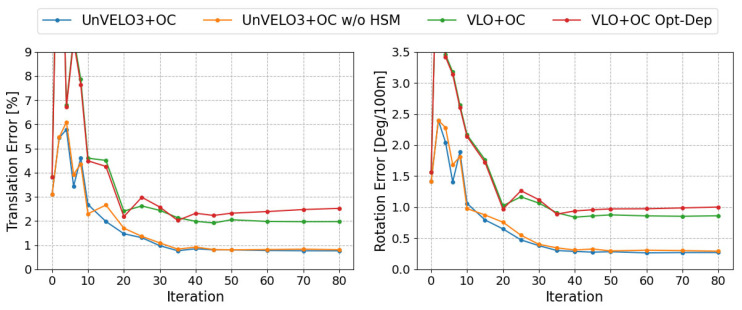
The performance of different model variants varied with respect to different iteration numbers. The translation and rotation errors are the mean errors of sequences 09 and 10.

**Figure 6 sensors-23-03967-f006:**
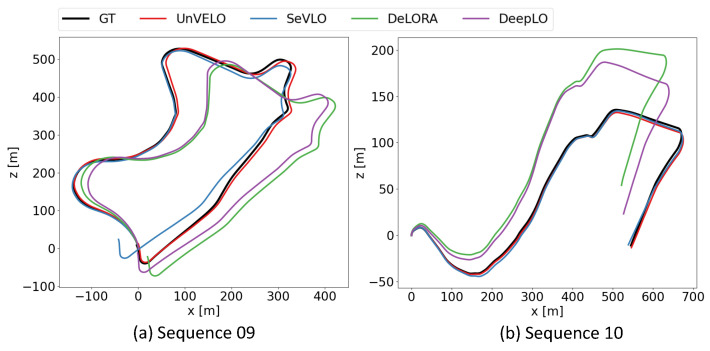
Trajectories of sequences 09 and 10 on KITTI.

**Figure 7 sensors-23-03967-f007:**
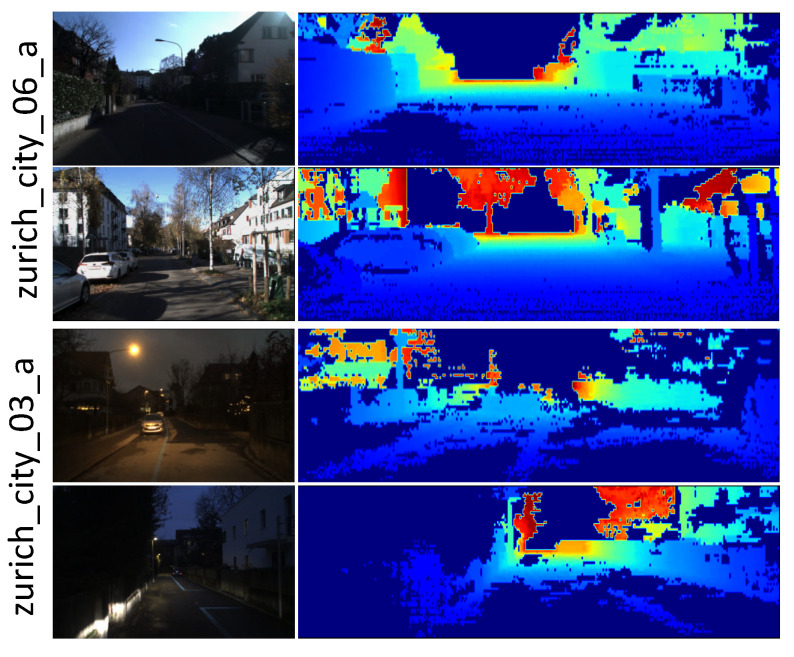
Typical samples in the test sequences of DSEC.

**Figure 8 sensors-23-03967-f008:**
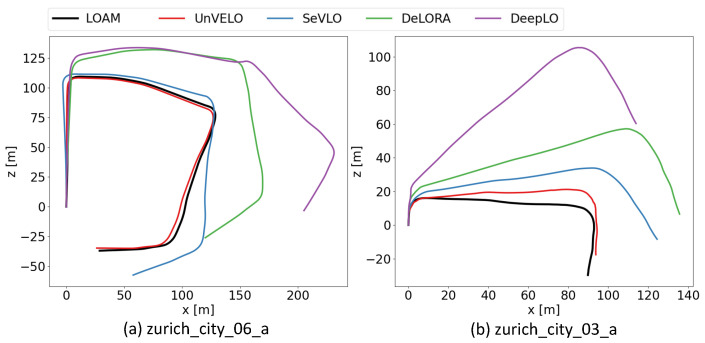
Trajectories of sequence “zurich_city_06_a” (day scenario) and “zurich_city_03_a” (night scenario) on DSEC.

**Table 1 sensors-23-03967-t001:** The performance of our pose-estimation network and its variants. The Modal column denotes the modalities of the network inputs, where “L” stands for LiDAR and “V” stands for visual. The best results are bold.

Models	Modal	Fusion Scheme	Loss	Seq.09		Seq.10	
				$t_{\mathrm{rel}}$	$r_{\mathrm{rel}}$	$t_{\mathrm{rel}}$	$r_{\mathrm{rel}}$
VLO [27]	L-Dep+V	VDF	-	4.33	1.72	3.30	**1.40**
UnVELO1	L-2D	-	$L_{geo}$	4.36	1.49	4.10	2.07
UnVELO2	L-2D+V	LDF	$L_{geo}$	3.83	1.29	4.10	1.86
UnVELO3	L-2D+V	LDF	$L_{geo}+L_{vis}$	**3.52**	**1.12**	**2.66**	1.71

“L-2D” and “L-Dep” denote the input LiDAR data are a 2D range map and sparse depth map, respectively.

**Table 2 sensors-23-03967-t002:** The performance and runtime of our UnVELO model with different numbers of iterations and the hard sample mining (HSM) scheme in online correction (OC). The best results are bold.

Models	Iter	Seq.09		Seq.10		Runtime (ms)
		$t_{\mathrm{rel}}$	$r_{\mathrm{rel}}$	$t_{\mathrm{rel}}$	$r_{\mathrm{rel}}$	
VLO	0	4.33	1.72	3.30	1.40	25.1
VLO+OC-10	10	3.69	1.91	5.51	2.41	219.6
VLO+OC-20	20	2.57	1.14	2.25	0.61	411.2
VLO+OC-40	40	2.58	1.02	1.40	0.66	705.3
VLO+OC-40 Opt-Dep	40	2.92	1.12	1.72	0.76	742.2
VLO+OC-40 w/o HSM	40	2.58	1.02	1.41	0.66	646.8
UnVELO3	0	3.52	1.12	2.66	1.71	20.1
UnVELO3+OC-10	10	2.10	0.94	3.26	1.18	142.1
UnVELO3+OC-20	20	1.53	0.53	1.42	0.77	245.4
UnVELO3+OC-40	40	0.99	**0.26**	0.71	0.31	458.6
UnVELO3+OC-40 w/o HSM	40	**0.96**	0.32	0.88	0.30	412.7
UnVELO3+OC-40-Inter2	40	1.04	0.27	**0.69**	**0.24**	-

**Table 3 sensors-23-03967-t003:** Comparison of the proposed method with the SoTA trained on Seq.00-08 and tested on Seq.09-10 of KITTI. The Modal column denotes the modalities of network inputs, where “L” stands for LiDAR and “V” stands for visual. The best results are bold.

	Method	Modal	Seq.09		Seq.10	
			$t_{\mathrm{rel}}$	$r_{\mathrm{rel}}$	$t_{\mathrm{rel}}$	$r_{\mathrm{rel}}$
Unsup.	DeepLO [18]	L-2D	4.87	1.95	5.02	1.83
	DeLORA [20]	L-2D	6.05	2.15	6.44	3.00
	SeVLO [27]	L-Dep+V	2.58	1.13	2.67	1.28
Hybrid	SS-DPC-Net [67]	V	2.13	0.80	3.48	1.38
	DeLORA w/ mapping [20]	L-2D	1.54	0.68	1.78	0.69
T-Opt.	DOC [15]	V	2.26	0.87	2.61	1.59
	DOC+ [15]	V	2.02	0.61	2.29	1.10
	Li et al. [58]	V	1.87	0.46	1.93	**0.30**
	Wagstaff et al. [66]	V	1.19	0.30	1.34	0.37
	UnVELO (Ours)	L-2D+V	**0.99**	**0.26**	**0.71**	0.31

“L-2D” and “L-Dep” denote the input LiDAR data are a 2D range map and sparse depth map, respectively.

**Table 4 sensors-23-03967-t004:** Comparison of the proposed method with the SoTA trained on Seq.00-06 and tested on Seq.07-10 of KITTI. The Modal column denotes the modalities of network inputs, where “L” stands for LiDAR and “V” stands for visual.

	Method	Modal	Seq.07		Seq.08		Seq.09		Seq.10	
			$t_{\mathrm{rel}}$	$r_{\mathrm{rel}}$	$t_{\mathrm{rel}}$	$r_{\mathrm{rel}}$	$t_{\mathrm{rel}}$	$r_{\mathrm{rel}}$	$t_{\mathrm{rel}}$	$r_{\mathrm{rel}}$
Sup.	PWCLO-Net [25]	L-P	0.60	0.44	1.26	0.55	0.79	0.35	1.69	0.62
	LO-Net [17]	L-2D	1.70	0.89	2.12	0.77	1.37	0.58	1.80	0.93
	E3DLO [21]	L-2D	0.46	0.38	1.14	0.41	0.78	0.33	0.80	0.46
Unsup.	UnPWC-SVDLO [46]	L-P	0.71	0.79	1.51	0.75	1.27	0.67	2.05	0.89
	SelfVoxeLO [24]	L-vox	3.09	1.81	3.16	1.14	3.01	1.14	3.48	1.11
	RLO [26]	L-vox	3.24	1.72	2.48	1.10	2.75	1.01	3.08	1.23
Hybrid	DMLO [45]	L-2D	0.73	0.48	1.08	0.42	1.10	0.61	1.12	0.64
	DMLO w/mapping [45]	L-2D	0.53	0.51	0.93	0.48	0.58	0.30	0.75	0.52
	LO-Net w/mapping [17]	L-2D	0.56	0.45	1.08	0.43	0.77	0.38	0.92	0.41
	SelfVoxeLO w/mapping [24]	L-vox	0.31	0.21	1.18	0.35	0.83	0.34	1.22	0.40
	RLO w/mapping [26]	L-vox	0.56	0.26	1.17	0.38	0.65	0.25	0.72	0.31
	UnVELO (Ours)	L-2D+V	1.46	0.78	1.25	0.43	0.88	0.26	0.79	0.33

“L-P” denotes the input LiDAR data are raw point clouds. “L-2D” denotes the input LiDAR data are a 2D range map, and “L-vox” denotes the input LiDAR data are 3D voxels.

**Table 5 sensors-23-03967-t005:** Evaluation results on the DSEC dataset. The Modal column denotes the modalities of network inputs, where “L” stands for LiDAR and “V” stands for visual. The best results are bold.

	Method	Modal	Day(06_a)		Night(03_a)	
			$t_{\mathrm{rel}}$	$r_{\mathrm{rel}}$	$t_{\mathrm{rel}}$	$r_{\mathrm{rel}}$
Unsup.	DeLORA [20] †	L-2D	22.53	9.93	39.00	12.48
	DeepLO [18] †	L-2D	42.43	20.77	51.26	31.12
	SeVLO [27]	L-Dep+V	8.42	5.78	22.43	24.37
	UnVELO (Ours)	L-2D+V	**2.07**	**2.10**	**5.84**	**7.92**

† The LiDAR only had 16 lines, which is too sparse for LOs. Therefore, LOs were also tested on 3D point clouds projected from disparity maps.

## Data Availability

The data presented in this study are available upon request. The datasets used in this study are openly available in https://www.cvlibs.net/datasets/kitti/eval_odometry.php (accessed on 1 May 2022) (KITTI odometry benchmark) and https://dsec.ifi.uzh.ch/dsec-datasets/download/ (accessed on 30 January 2023) (DSEC dataset).

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
