# Peer review of "UnVELO: Unsupervised Vision-Enhanced LiDAR Odometry with Online Correction"

_sensors, 2023, doi:10.3390/s23083967_

Round 1

Reviewer 1 Report

This study proposed a new method to implement an unsupervised VLO, which adopts a LiDAR-dominant scheme to fuse two modalities. The method in this study was tested on KITTI and DSEC datasets. The proposed method outperformed previous two-frame based learning methods. It is interesting and helpful for this topic research and future study.

However, there are some issues or questions could be addressed:

1) In the abstract, the meaning and the important findings of this study should be explained clearly. The meaning of the proposed method is not only the comparitive performance.

2) In the Section of Introduction and Related work, the existing methods such as learning-based vision-dominant and LiDAR frame odometric were described. But, the problems of previous studies are not described. There is not the explanation of the detailed problems or limitations in these methods. 

3) The Figure 1 should be placed in Section 3.

4) In Section 3.3 of Pose Online Correction, what is the meaning or role of 3.3.3 Hard Sample Mining, which was also not represented in Figure 1.

5) In Section 4 of Results, this important results should be represented in summary. 

6) In Second 5 of Discussion, the content of this section is conclusion in fact. The title of this section should be revised. So, the discussion of this study need to be further explored. 

Author Response

First of all, we would like to thank all reviewers and editors for your constructive comments and the time you dedicated to our manuscript. After considering all comments carefully, we revised the manuscript as required by adding more clarifications and results analysis. All changes in our manuscript are highlighted in blue.

For the point-py-point response, please see the attachment.

Reviewer 2 Report

Lidar odometry is a very interesting topic in the research area. This work proposes a new method to implement an unsupervised vision-enhanced LiDAR odometry (UnVELO).The paper is well organized and written, the experiments are also sufficient. However, the paper could be further improved by considering the following aspects:

1. Currently, the research in this area is mainly based on multi-sensors, such as IMU, GNSS, camera, and LiDAR. There is a lot of related work on how to process the sensor data, how to consider the motion states etc. Please elaborate more on the literature review section from the configuration of the sensor: improved vehicle localization using on-board sensors and vehicle lateral velocity.

2. in figure 7, the units of the axes should be added

3. a conclusion section should be included in the paper, instead of the “Discussion” at the end of the paper.

Author Response

(The authors gave the same response as above.)
